# The Importance of Skeletal Muscle Oxygenation Kinetics for Repeated Wingate-Based Sprint Performance

**DOI:** 10.3390/muscles4020018

**Published:** 2025-06-13

**Authors:** Andrew Usher, John Babraj

**Affiliations:** Department of Sport and Exercise Science, Abertay University, Bell St., Dundee DD1 1HG, UK; j.babraj@abertay.ac.uk

**Keywords:** NIRS, rectus femoris, oxygenation, Wingate, recovery

## Abstract

Boxing is a sport that has a high level of oxygen use within the Rectus Femoris muscle, with recovery between rounds important to subsequent performance. The study aimed to determine muscle oxygen use in male and female professional boxers in response to a repeated sprint stimulus. 10 male (age: 26 ± 5 years, height: 177 ± 4 cm, weight: 72 ± 6 kg) and 6 female (age: 29 ± 4 years, height: 173 ± 2 cm, weight: 73 ± 4 kg) professional boxers took part. Participants attended a single session where a Moxy near-infrared monitor was placed on the rectus femoris muscle of both legs. Participants completed 2 × 30 s Wingate-based sprint efforts with a 60 s active recovery (maintaining 60 rpm against 50 W resistance). Skeletal muscle oxygenation was recorded throughout. Significant differences were found in peak power, average speed and rate of fast desaturation between males and females (*p* < 0.001) in both sprints. There was a significant difference in males and females for the rate (sprint 1M: −5.6 ± 1.2%.s^−1^, 1F: −2.3 ± 1.9%.s^−1^, *p* < 0.05; sprint 2M: −4.2 ± 1.1%.s^−1^, 2F: −1.5 ± 0.9%.s^−1^, *p* < 0.05) and duration of fast desaturation (sprint 1M: 6.1 ± 1.3 s, 1F: 3.7 ± 2.8 s, *p* < 0.05; sprint 2M: 7.3 ± 1.6 s, 2F: 4.5 ± 1.0 s, *p* < 0.05) in both sprints. This study demonstrates that male professional boxers have a significantly different oxygen use and recovery in response to a high-intensity stimulus compared to female boxers. In both male and female boxers, the ability to perform subsequent high-intensity activity is dependent on the quality of the recovery from the initial high intensity activity. Therefore, there is a greater need to focus training to improve recovery kinetics in boxing.

## 1. Introduction

Boxing is a highly technical sport [1], with athletes competing over varying weight classifications and round formats (amateur boxing 3 rounds, professional, 4–12). It requires a combination of relevant strength and conditioning [2], multidirectional footwork, evasion skills, striking ability and tactical awareness. Boxing can be classified as an intermittent high-intensity sport, with the unique feature of 1 min break at the end of each round. Research to date has mainly focused on male amateur boxing and has shown the kinematic sequence of striking [3] with EMG data showing peak activation within the rectus femoris muscle within the initiation phase of the jab, cross and rear hand hook punch. Since the rectus femoris muscle is crucial to punch force generation then there is a need to explore the physiological response of this muscle in boxers. In male professional boxers, we have previously shown a decline in muscle oxygen response of the rectus femoris muscle across multiple rounds of sparring [4], which may limit the athlete’s ability to perform effective punch technique in later rounds. However, the extent of the metabolic disturbance may be affected by the quality of the opposition [1].

To date, the energetic costs of boxing have been estimated to be either 80% anaerobic energy use and 20% aerobic energy use [5], or to be 80% aerobic energy and 20% anaerobic energy [6]. This may reflect differences in the work rate of the bouts, with Ghosh having an activity frequency of 1.7–2.0/s and Davies having an activity frequency of 1.2/s. Regardless of energetic breakdown, there is a crucial role for anaerobic energy production during a boxing match. Wingate performance is viewed as a gold standard for anaerobic testing, with peak power linked to phosphocreatine breakdown and mean power to anaerobic glycolysis and lactate production [7,8]. In male amateur boxers, lower body Wingate performance has been shown to be between 9.8 and 11.8 W·kg^−1^ for peak power and 6.7–9.5 W·kg^−1^ for mean power [9]. In female amateur boxers, lower body Wingate peak power has been reported as 7.6 W·kg^−1^ and mean power as 6.3 W·kg^−1^ [10]. For male amateur boxers, upper body Wingate performance has been shown to be 8.4 W·kg^−1^ for peak power and 6.3 W·kg^−1^ for mean power [11]. The value for lower body and upper body Wingate performance in male and female national team boxers is lower than that seen in combat sports such as wrestling [11,12] or judo [13]. However, power may not be a good metric of muscular demand during a Wingate test due to the inefficiency of transfer of energy from the pedal to the flywheel [14]. The kinetic energy produced in the muscle during a cycle stroke is directly related to the pedal rate [15]. This may mean comparison across studies is incorrect due to differences in ergometer mechanics and cadence metrics should also be reported to allow for better comparison. This means there is a need to explore more fully the nature of cycling dynamics during this test to gain a better understanding of an athlete’s performance.

Whilst the Wingate test is thought to be anaerobic, near-infrared spectroscopy has been utilised to look at muscular oxygen demand during it. In club track and field athletes, vastus lateralis oxygenation has been shown to fall by 17% during the Wingate test [16], with the dynamics of oxygen recovery being slower in long-distance runners compared to sprinters. In elite handball players, there was a greater oxygen recovery in the triceps brachii muscle after an upper body Wingate test compared to control participants, although the total fall in muscle oxygenation was similar between groups [17]. Recently, it has been suggested that the muscle oxygen response follows a monoexponential decay, with an initial time delay of 2.9 s before the vastus lateralis begins to utilise oxygen [18]. However, the dynamic nature of muscle oxygenation during repeated Wingate-based sprints and recovery has not been explored.

Given the high-intensity nature of boxing competition with 1 min recovery between rounds then the use of repeated Wingate-based sprints may allow an accurate reflection of how well a boxer will perform and recover. In soccer players, repeated Wingate-based sprints have been shown to result in a 13% decrease in peak power and a 30% decrease in mean power from sprint 1 to sprint 2 with 15 s recovery between sprints [19]. In male judokas, repeated 30 s Wingate-based sprints result in peak power decreasing by 13% and mean power by 16% with 4 min recovery between sprints [13]. In female judokas, repeated 30 s Wingate-based sprints result in peak power decreasing by 10% and mean power by 13% with 4 min recovery between sprints [13]. There was a greater reduction in vastus lateralis oxygen saturation in the male judokas compared to the female judokas during each repeated sprint, with a similar oxygenation profile throughout when 4 min of recovery are given [13]. To date, there is no exploration of repeated Wingate-based sprint performance in amateur or professional boxers or of the rectus femoris muscle. Therefore, the aim of the current study is to determine repeated Wingate-based sprint performance and rectus femoris muscle oxygenation with 1 min of recovery between sprints in male and female professional boxers. It was hypothesised that there would be a decrease in power across the 2 sprint efforts and this would be related to the ability of the rectus femoris muscle to recover from the initial sprint effort. Further, it was hypothesised that the response to repeated Wingate-based sprint efforts would be different in male and female boxers.

## 2. Methods

Participants: 10 male professional boxers (age: 26 ± 5 years, height: 177 ± 4 cm, weight: 72 ± 6 kg, Body Fat: 10 ± 5%) and 6 female professional boxers (age: 29 ± 4 years, height: 173 ± 2 cm, weight: 73 ± 4 kg, Body Fat 22 ± 4) were recruited for this study. This represents 10% of female professional boxers in the UK and 1% of male professional boxers in the UK. Each boxer held a current professional licence with the British Boxing Board of Control. Participants were excluded from the study if they had any loss of training days over the last 3 months due to musculoskeletal injury or sanction from the British Boxing Board of Control preventing the athlete from taking part in sparring. All participants recruited were in training camps for upcoming fights but were at different stages of preparation. Prior to the study, participants were informed both verbally and in writing on the risks and benefits of the study before giving informed consent. The study was approved by Abertay University Ethics Committee (EMS7056 and EMS4768) and carried out in line with the Declaration of Helsinki, except for the registration in a database.

No attempt was made to control for menstrual cycle or oral contraceptive use in the female participants. It has been demonstrated during exercise that the menstrual phase or oral contraceptives have no impact on the NIRS signal [20]. Prior to the session, athletes were advised to fast for 2 h, with no food or water consumed. Upon arrival at the laboratory, the participants had near-infrared spectrometers (Moxy monitor, Fortiori Design LLC, Hutchison, MN, USA) attached to the rectus femoris muscle of the left and right leg. The monitor was set to acquire data at a frequency of 2 Hz with no smoothing. The rectus femoris muscle was identified by palpation during an isometric contraction and the moxy monitor was then taped to the belly of the muscle. Data for the left and right leg were then collected via Bluetooth transmission to the VO2 master app version 0.50.0 (VO2 Master Health Sensors Inc., Vernon, BC, Canada).

Lower body repeated Wingate-based sprints: Seat height was adjusted on the electronically braked ergometer (Lode Excalibur Sport, Lode BV, Groningen, The Netherlands) to ensure the participant had full leg extension. Participants then cycled for 60 s against a load of 50 W prior to the start of the test and were told to try and hit maximum speed as fast as possible when they were told to go. Also, 30 s prior to the start of the Wingate the participants brought the cadence to 85 rpm and were instructed to increase the cadence during the 3, 2, 1 countdown to the start of the test. Participants then cycled for 30 s against an electronically braked resistance equivalent to 7.5% body mass (Torque factor 0.75 N·m·kg·bodymass^−1^). There was then a 60 s active recovery cycling against a load of 50 W prior to a second 30 s Wingate effort carried out against the same resistance. For each Wingate sprint, peak power, average power, minimum power and time to peak were collected.

### 2.1. Data Analysis

SmO_2_ was used for data analysis as it gives a better reflection of muscle oxygenation when blood flow is not in a steady state [21]. SmO_2_ data was exported from the VO2 master as a 1 s average before being processed as previously described [4]. Briefly, a 5 s median filter was applied to the data to remove movement artefacts prior to linear curve fitting being applied to the data to look at the relationship between SmO_2_ and time and SmO_2_ and heart rate. Linear segments were identified based on the point of rapid SmO_2_ desaturation, typically occurring near peak power output. Segments were selected using both visual inspection and confirmation via high R^2^ values (>0.90) from the linear regression fit. Three clear components in the NIRS signal are present in each sprint: initial time delay where oxygenation is stable, a fast desaturation where oxygenation levels are falling and a new steady state where oxygenation is lower but stable (Figure 1). The new steady state was defined as the end of the fast linear desaturation until the end of the 30 s period. Two clear linear components are present during recovery: initial time delay where oxygenation is stable, a fast resaturation where oxygenation levels are rising rapidly before going into a non-linear component (Figure 1). Time positive acceleration was calculated using a monoexponential fit to the Lode acceleration across the Wingate sprint as follows:Acc (t) = AM + A × exp^(−(x − TD/t))^
where Acc (t) is the acceleration at a given time, AM is the max acceleration, A is the amplitude of change, TD is the time delay and t is the time constant to plateau.

### 2.2. Statistical Analysis

All data are reported as means ± standard deviation. All statistical analysis was carried out using Jamovi (version 2.3.13). Data was checked for normality using the Shapiro–Wilks test for normality and all data was deemed to be normally distributed. All data was analysed using a repeated measures ANOVA for sprints separated by sex. Where there was a significant main effect or interaction effect for sprint, leg or sex then a least squares difference post hoc test was applied to determine where differences occurred. Significance was accepted at *p* < 0.05. The multivariate correlation was carried out to determine if there was a relationship between Wingate-based sprint speed and power components and left and right leg rectus femoris desaturation or resaturation components. Hedges G effect size was calculated for all variables and presented with 95% confidence intervals. If the 95% confidence interval crossed zero, then the effect was deemed unreliable. Magnitude of effect was defined as <0.25 trivial effect, 0.25–0.5 small effect, 0.5–1.0 moderate effect and >1.0 as a large effect [22].

## 3. Results

### 3.1. Lower Body Wingate Power

There was no significant main effect for a difference in time to peak power between males and females (with a small to moderate unreliable effect; *p* > 0.05; Figure 2) or across sprints (with a small to moderate unreliable effect; *p* > 0.05; Table 1). There was a significant main effect for sprint (*p* < 0.001; Table 1) and sex (*p* < 0.001; Table 1), with a difference in peak power between males and females for sprint 1 and 2 (with a large reliable effect; *p* < 0.001; Table 1) and between sprint 1 and sprint 2 (with a large reliable effect; *p* < 0.001; Table 1). When peak power was normalised to fat-free mass, there was a significant main effect for sprint (*p* < 0.001; Table 1) and sex (*p* = 0.018; Table 1), with a difference between sprint 1 and 2 for both males and females (with a large reliable effect; *p* < 0.001; Table 1) and a significant difference only in sprint 2 between the sexes (with a large reliable effect for sprint 2 and a moderate unreliable effect for sprint 1; *p* = 0.010; Table 1). There was a significant main effect for sprint (*p* < 0.001; Table 1) and sex (*p* < 0.001; Table 1), with a difference in average power between males and females (with a large reliable effect; *p* < 0.001; Table 1) and between sprint 1 and sprint 2 for both males and females (with a large reliable effect; *p* < 0.001; Table 1). When average power was normalised to fat-free mass, there was a significant main effect for sprint (*p* < 0.001; Table 1) and sex (*p* = 0.007; Table 1), with a difference between males and females (with a large reliable effect; *p* < 0.02; Table 1) and between sprint 1 and sprint 2 for both males and females (with a large reliable effect; *p* < 0.001; Table 1). There was a significant main effect for sprint (*p* < 0.001; Table 1), with a significant difference between sprint 1 and sprint 2 for both males and females (with a large reliable effect for females but a moderate unreliable effect for males; *p* < 0.001; Table 1) but no significant difference in minimum power between males and females (with a small to moderate unreliable effect; *p* > 0.05; Table 1). When minimum power was normalised to fat-free mass, there was a significant main effect for sprint (*p* < 0.001; Table 1), with a significant difference between sprint 1 and 2 for both males and females (with a large reliable effect for females but a moderate unreliable effect for males; *p* < 0.001; Table 1) but no significant difference between the sexes (with a small to moderate unreliable effect; *p* > 0.05; Table 1).

### 3.2. Lower Body Wingate Speed

There was no significant main effect for the difference in time of positive acceleration (Figure 3) between males and females or across sprints (with a small unreliable effect; *p* > 0.05; Table 1). There was a significant main effect for the sprint for max speed (*p* < 0.001; Table 1), with a significant difference in max speed across sprints for males but not females (with a large reliable effect for males but a moderate unreliable effect for females; *p* < 0.001; Table 1) but no significant difference between males and females (with a trivial to small unreliable effect; *p* = 0.260, Table 1). There was a significant main effect for sprint (*p* < 0.001; Table 1) and sex (*p* < 0.001; Table 1), with a difference in average speed between males and females (with a large reliable effect; *p* < 0.001; Table 1) and between sprint 1 and sprint 2 (with a large reliable effect; *p* < 0.001; Table 1). There was a significant main effect for sex (*p* < 0.001; Table 1), with a significant difference between males and females (with a large reliable effect; *p* < 0.02; Table 1) but no significant difference in time to max speed across sprints (with an unreliable effect; *p* = 0.343, Table 1). There was a significant main effect for sprint (*p* = 0.04; Table 1) and sex (*p* < 0.001; Table 1), with a difference in time to speed loss across sprints 1 and 2 for males but not females (with a small to moderate unreliable effect; *p* = 0.040, Table 1) and there was a significant difference between males and females (with a large reliable effect; *p* < 0.025; Table 1). There was a significant main effect for sprint (*p* = 0.02; Table 1) and sex (*p* = 0.024; Table 1), with a significant difference in time at max speed across sprints for males (with a large reliable effect for males but a trivial unreliable effect for females; *p* < 0.001, Table 1) and there was a significant difference between males and females for sprint 1 but not sprint 2 (with a large reliable effect for sprint 1 but a trivial unreliable effect for sprint 2; *p* = 0.009; Table 1).

### 3.3. Rectus Femoris Muscle Oxygenation

There was no significant main effect for the difference in muscle oxygenation characteristics between the left and right rectus femoris muscle (*p* < 0.05; Table 2). There was no significant main effect for the difference in the time delay to fast desaturation between males and females (with a trivial unreliable effect; *p* > 0.05; Table 2; Figure 1) or across sprints (with a trivial unreliable effect; *p* > 0.05; Table 2; Figure 1) for either left or right leg. There was a significant main effect for sprint (*p* < 0.001; Table 2) and sex (*p* < 0.001; Table 2), with a difference in the rate of fast desaturation between males and females in both sprints (with a large reliable effect; *p* < 0.005; Table 2; Figure 1) and between sprint 1 and sprint 2 for males (with a large reliable effect for males and a moderate unreliable effect for females; *p* < 0.05; Table 2; Figure 1) for the left leg. There was a significant difference in the rate of fast desaturation between males and females (with a large reliable effect for sprint 1 and a large unreliable effect for sprint 2; *p* < 0.005; Table 2; Figure 1) and between sprint 1 and sprint 2 for males but not females (with a small unreliable effect for males and a trivial unreliable effect for females; *p* < 0.05; Table 2; Figure 1) for the right leg. There was a significant main effect for sex (*p* < 0.001; Table 2) with difference in the duration of the fast desaturation between males and females for sprint 1 and 2 (with a large reliable effect; *p* < 0.01; Table 2; Figure 1) but no significant difference between sprint 1 and sprint 2 (with a trivial to moderate unreliable effect; *p* > 0.05; Table 2; Figure 1) for either right or left leg. During the 60 s recovery, there was no significant difference in the time delay to fast resaturation (with a trivial to small unreliable effect; *p* > 0.05; Table 2; Figure 1) for either the right or left leg. There was a significant main effect for sex (*p* = 0.005; Table 2) with a difference in the rate of fast resaturation between males and females for sprint 2 but not for sprint 1 (with a large reliable effect for sprint 2 and a moderate unreliable effect for sprint 1; *p* < 0.001; Table 2; Figure 1) but no significant difference for sprint (with a small unreliable effect; *p* < 0.05; Table 2) for the left leg. There was no significant difference in the rate of fast resaturation between males and females (with a moderate to large unreliable effect; *p* > 0.05; Table 2; Figure 1) or for sprint (with a small unreliable effect; *p* < 0.05; Table 2) for the right leg. There was a significant main effect for sex (*p* = 0.028; Table 2), with a difference in the time duration of fast resaturation between males and females for sprint 1 but not sprint 2 (with a large unreliable effect for sprint 1 and a moderate unreliable effect for sprint 2; *p* = 0.018 for sprint 1; Table 2; Figure 1) but no significant effect for sprint (*p* > 0.05; Table 2) in the right leg. There was no significant difference for sex (with a trivial to moderate unreliable effect; *p* > 0.05; Table 2) or for sprint (with a small unreliable effect; *p* > 0.05; Table 2) in the left leg.

There were no significant main or interaction effects for starting SmO_2_ during the repeated Wingate-based sprints (with a trivial to moderate unreliable effect, *p* > 0.05, Table 2, Figure 1) for both the left and right leg. There was a significant main effect of sex (*p* = 0.010) and a sex-bysprint interaction effect (*p* = 0.001) for steady-state SmO_2_ during the repeated Wingate-based sprints (Table 2, Figure 1). There were no significant differences for steady state SmO_2_ in males or females by leg (*p* > 0.05, Table 2). There was a significant difference in steady state SmO_2_ for males compared to females in sprint 1 for right (with a large reliable effect; *p* = 0.005, Table 2) and left (with a large reliable effect; *p* = 0.010, Table 2) leg and in sprint 2 for right leg (with a large reliable effect; *p* = 0.041, Table 2) but not left leg (with a large reliable effect; *p* = 0.068, Table 2). For males only, there was a significant difference in steady state SmO_2_ for sprint 1 compared to sprint 2 in the right (with a small unreliable effect; *p* < 0.001, Table 2) and left (with a moderate unreliable effect; *p* = 0.004, Table 2) leg.

### 3.4. Multivariate Correlative Analysis

Left leg and Sprint 1: Multivariate regression analysis shows that left leg skeletal muscle oxygenation components explain 50% of the time for positive acceleration with significant association for the time delay before desaturation (coefficient: 0.26 [95%CI: 0.05:0.47]; *p* = 0.021), 73% of the time at max speed with significant association for the rate of fast desaturation (coefficient: 0.57 [95%CI: 0.07:1.11]; *p* = 0.028) and the steady state SmO2 (coefficient: −0.11 [95%CI: −0.03:−0.18]; *p* = 0.008), 69% of the time to speed loss with significant association for the fast desaturation duration (coefficient: 0.45 [95%CI: 0.01:0.89]; *p* = 0.046), 43% of the time to peak power with significant association for the time delay for fast desaturation (coefficient: 0.14 [95%CI: 0.01:0.26]; *p* = 0.033). There were no significant associations between left leg skeletal muscle oxygen components and other components of sprint performance (*p* > 0.05).

Right leg and Sprint 1: Multivariate regression analysis shows that right leg skeletal muscle oxygenation components explain 49% of max speed with significant association for the time delay to fast desaturation (coefficient: 2.65 [95%CI: 0.24:4.87]; *p* = 0.024). There were no significant associations between right leg skeletal muscle oxygen components and other components of sprint performance (*p* > 0.05).

Left leg and Sprint 2: Multivariate regression analysis shows that left leg skeletal muscle oxygenation components explain 52% of the time for positive acceleration with significant association for the time delay before desaturation (coefficient: 0.28 [95%CI: 0.06:0.51]; *p* = 0.019), 57% of the time at max speed with significant association for the rate of fast desaturation (coefficient: 0.38 [95%CI: 0.06:0.69]; *p* = 0.025) and the steady stateSmO_2_ (coefficient: −0.05 [95%CI: −0.01:−0.09]; *p* = 0.030), 64% of the time to speed loss with significant association for the fast desaturation duration (coefficient: 0.68 [95%CI: 0.02:1.40]; *p* = 0.045), 87% of the mean power with significant association for the fast desaturation duration (coefficient: 0.28 [95%CI: 0.05:0.51]; *p* = 0.020), 52% of the minimum power with significant association for the starting SmO_2_ (coefficient: 0.11 [95%CI: 0.01:0.20]; *p* = 0.045), 72% of the mean power normalised to lean mass with significant association for the fast desaturation duration (coefficient: 0.32 [95%CI: 0.01:0.63]; *p* = 0.043). There were no significant associations between left leg skeletal muscle oxygen components and other components of sprint performance (*p* > 0.05).

Right leg and Sprint 2: Multivariate regression analysis shows that right leg skeletal muscle oxygenation components explain 39% of the time for positive acceleration with significant association for the time delay before desaturation (coefficient: 0.26 [95%CI: 0.01:0.52]; *p* = 0.045), 45% of the time at max speed with significant association for the rate of fast desaturation (coefficient: 0.20 [95%CI: 0.01:0.40]; *p* = 0.047) and the steady stateSmO_2_ (coefficient: −0.06 [95%CI: −0.01:−0.12]; *p* = 0.031), 41% of the minimum power with significant association for the steady stateSmO_2_ (coefficient: −0.16 [95%CI: −0.01:−0.30]; *p* = 0.041), 53% of the min power normalised to lean mass with significant association for the starting SmO_2_ (coefficient: 0.15 [95%CI: 0.02:0.27]; *p* = 0.026) and the steady stateSmO_2_ (coefficient: −0.17 [95%CI: −0.03:−0.30]; *p* = 0.020). There were no significant associations between right leg skeletal muscle oxygen components and other components of sprint performance (*p* > 0.05).

Left leg sprint 1 recovery and sprint 2 performance: Multivariate regression analysis shows that left leg skeletal muscle oxygenation components during recovery from sprint 1 explain 68% of average speed during sprint 2, with a significant relationship with fast resaturation rate (coefficient: 4.64 [95%CI: 0.25:9.03]; *p* = 0.018), fast resaturation delay (coefficient: 1.62 [95%CI: 0.21:3.03]; *p* = 0.028) and steady stateSmO2 sprint 1 (coefficient: −0.28 [95%CI: −0.01:−0.55]; *p* = 0.041), 54% of time to speed loss during sprint 2, with a significant relationship with resaturation duration (coefficient: 0.34 [95%CI: 0.04: 0.65]; *p* = 0.032), 74% of mean power during sprint 2, with a significant relationship with resaturation duration (coefficient: 0.19 [95%CI: 0.05:0.32]; *p* = 0.012) and steady stateSmO_2_ sprint 1 (coefficient: −0.04 [95%CI: −0.01:−0.07]; *p* = 0.006), 82% of mean power normalised to lean body mass during sprint 2, with a significant relationship with time delay to fast resaturation (coefficient: 0.04 [95%CI: 0.01:0.07]; *p* = 0.025), fast resaturation duration (coefficient: 0.23 [95%CI: 0.12:0.33]; *p* = 0.001) and steady stateSmO_2_ sprint 1 (coefficient: −0.04 [95%CI: −0.02:−0.06]; *p* = 0.002), 53% of min power normalised to lean body mass during sprint 2, with a significant relationship with fast resaturation duration (coefficient: 0.24 [95%CI: 0.05:0.44]; *p* = 0.020), 52% of peak power normalised to lean body mass during sprint 2, with a significant relationship with fast resaturation duration (coefficient: 0.23 [95%CI: 0.02:0.46]; *p* = 0.038). There were no significant associations between left leg skeletal muscle oxygenation components during recovery and other components of sprint 2 performance (*p* > 0.05).

Right leg sprint 1 recovery and sprint 2 performance: There were no significant associations between right leg skeletal muscle oxygenation components during recovery and any components of sprint 2 performance (*p* > 0.05).

## 4. Discussion

### 4.1. Power

For lower body repeated Wingate sprints, we report a lower relative peak and average power (Figure 2) in females compared to males across both sprints (Table 1), even though body mass was matched. The values for male peak power are similar to those reported for elite male amateur boxers [11,23] but the values for female peak power are greater than those reported for elite female amateur boxers [10]. The sex difference reported is similar to the difference reported by others [24,25,26] where the females had a lower body mass by at least 10 kg. When power was normalised to fat-free mass there was still a significant difference in average power production across both sprints and peak power production in the second sprint (Table 1). This is different to what has been found previously where power recovery was similar in males and females [27]. In elite cyclists, with similar endurance capacity, power differences are not present across sex when normalised to fat-free mass across five short sprints or greater peak power in females during a 30 s sprint [28]. This suggests that there is a poorer endurance development in female boxers than in male boxers. However, given that power recovery is an aerobic process [29,30] then both male and female boxers current training regimes do not target endurance aspects to optimise lower body power recovery. This means there will be a loss of performance in professional boxers between rounds leading to an inability to maintain power production across subsequent rounds.

### 4.2. Speed

Despite the differences in power, the maximum speed (Figure 3) achieved during both Wingates is similar for males and females (Table 1), as is the time spent in positive acceleration (Table 1). However, there was a significant difference in the time taken to reach maximum speed, with males taking longer to reach maximum speed in both Wingates compared to females (Table 1). The similar maximum speed between male and female boxers is comparable to what has been reported previously for recreational athletes [31]. Power in cycling is a relationship between optimal velocity and optimal force, and optimal velocity is influenced by speed [15]. So, the speed developed and the duration of maximum speed will reflect fibre recruitment pattern within the skeletal muscle [32]. In both Wingate-based sprints, there is a significant association between speed metrics and left and right leg skeletal muscle oxygenation pattern, with power only becoming associated during the second Wingate-based sprint. Therefore, the maximum speed and the time components within this may tell us more about the performance of an athlete than simple power metrics based on flywheel inertia, although more work is required.

### 4.3. Oxygenation

During lower body sprints the time delay for rectus femoris muscle desaturation is the same in males and females and between sprints 1 and 2 (Table 2, Figure 1). This is the same value as reported for the exponential fit of SmO_2_ data during a Wingate test [18] and reflects a point where close to maximum power has been reached (Table 1) yet the amount of oxygen around the rectus femoris muscle is unchanged despite the increased work. Given this is the only time within the test where muscle oxygen levels are unchanged from resting, then it is the only part of the test that is predominantly anaerobic in nature. This is in line with systemic oxygen consumption which is increased during the first 5 s of the Wingate test [33].

There is then a significant difference between males and females in the rate of fast desaturation in each sprint (Table 2, Figure 1), with a much slower rate of desaturation in females than males. This reflects the rate that mitochondrial activity is exceeding the ability to supply oxygen to the muscle leading to decreased oxygenated myoglobin in the muscle [4] suggesting a slower rate of mitochondrial activation in females. This in contrast to what has previously been reported for desaturation during a graded exercise test at the second lactate threshold, where no difference in desaturation slope is reported in the vastus lateralis muscle for male and female trained cyclists [34]. The rate of desaturation is lower than that reported previously of −8.4%.s^−1^ in the vastus lateralis in endurance-trained males (VO_2_ max 55 mL.min^−1^.kg^−1^) [35]. In males, there was a significant reduction in the rate of desaturation from sprint 1 to sprint 2 but was compensated for by a longer duration of desaturation (Table 2). There will be a greater microcirculatory flow between sprints 1 and 2 for males compared to females [36]; however, if this was affecting response then you would expect a greater starting SmO_2_ in sprint 2 in males, but we do not see this (Table 1). Therefore, it is more likely to reflect the metabolic disturbance in the mitochondria, preventing a more rapid switch on of mitochondrial enzyme activity [37]. In support of this, we see a higher steady state SmO_2_ in males for sprint 2 compared to sprint 1 (Table 1), which suggests a loss of oxygen use in the skeletal muscle of males during repeated sprint activity that is not present in females (Table 1).

The rate of linear fast resaturation is much lower than the rate of linear fast desaturation during exercise (Table 2) but has a similar duration. The response then goes into a non-linear slower recovery phase before returning to baseline (Figure 1). There is a significantly faster linear resaturation rate in males compared to females for the left rectus femoris muscle for sprint 2 but not for sprint 1 or for the right leg. This may limit the speed developed in the second sprint, with a significant correlation between left leg fast resaturation, resaturation delay and end SmO_2_ from sprint 1 and average speed. Indeed, we show significant associations with a number of the Wingate sprint performance metrics and the recovery kinetics for the left leg from sprint 1. However, these are not present in the right leg. In sprint 2 the skeletal muscle oxygen components of the left leg are significantly associated with both speed and power metrics of the Wingate-based sprint. This suggests a greater reliance on oxidative fuel in the second sprint as shown previously through biopsy [38]. To date, no other study has looked at recovery kinetics following Wingate-based exercise and how it impacts subsequent performance. More exploration is needed to determine the impact of these recovery dynamics and leg differences on performance in boxing and its response to training interventions.

## 5. Limitations

There was no attempt to control for training load in the boxers, availability rather than a stage of fight camp determined participant recruitment, although all boxers were involved at some stage of fight preparation. This may mean the variation in the data may reflect the athlete’s stage of training, preventing the detection of small differences [39]. The overall sample size is small, although does represent 10% of all UK-registered female professional boxers and may limit multivariate regression analysis whereby associations between metrics are not identified. It may also increase the risk of type II errors leading to small differences between male and female boxers being missed. However, we do report robust associations across a number of variables and effect size that are reported to add weight to statistical findings in the current study. We have used the Lode ergometer for lower body Wingate and there are differences in power calculation between the Lode and the Monark ergometer [40], therefore the absolute comparisons between published data may need to be viewed in this light. Despite this, the trends seen between males and females and between sprints 1 and 2 would still be present with either bike. While we see no significant difference between the left and right leg for muscle oxygenation there are subtle differences between them (Table 2), which may be related to preferred stance. Small differences have been reported previously between dominant and non-dominant legs in team sports [41] but this is not seen in submaximal cycling [42]. Further research is needed to explore the importance of oxygenation asymmetries in elite athletes.

## 6. Perspective

We demonstrate for the first time the kinetic response of skeletal muscle oxygenation in male and female boxers during repeated Wingate-based sprints with a typical recovery period as seen between boxing rounds. The use of repeated Wingate tests is a powerful tool to explore the decline in muscle capacity and oxygenation and allow a greater understanding of an athlete’s ability to recover. This could be a useful tool within boxing to assess overall athlete readiness and ability to recover during rounds. Speed and acceleration may offer more information on Wingate performance than just reporting power calculated from the flywheel and as such should be reported in studies using the Wingate-based sprinting protocols. This, together with the extent of oxygen disturbance in the skeletal muscle suggests that an appreciable aerobic load occurs rapidly during Wingate-based sprints, and the extent of the anaerobic component may be as short as the time delay of muscle desaturation (Table 2) and this is similar for males and females. The rates of oxygen use between male and female boxers are different and this may represent a deficit in adaptation to traditional boxing training approaches in females and there may be a need to train females differently than males. However, for both males and females, there is poor recovery from Wingate-based sprints which impacts on the ability to create force in the muscle. This suggests there is a need for better conditioning in professional boxing. For coaches and trainers, the application of NIRS technology into testing allows an in-depth exploration of muscle function. When using with repeated Wingate-based sprints they should look at the delay to desaturation and rate of desaturation and how reproducible it is across efforts. In terms of recovery the rapid linear resaturation period is crucial to understanding an athlete’s ability to recover across rounds.

## Figures and Tables

**Figure 1 muscles-04-00018-f001:**
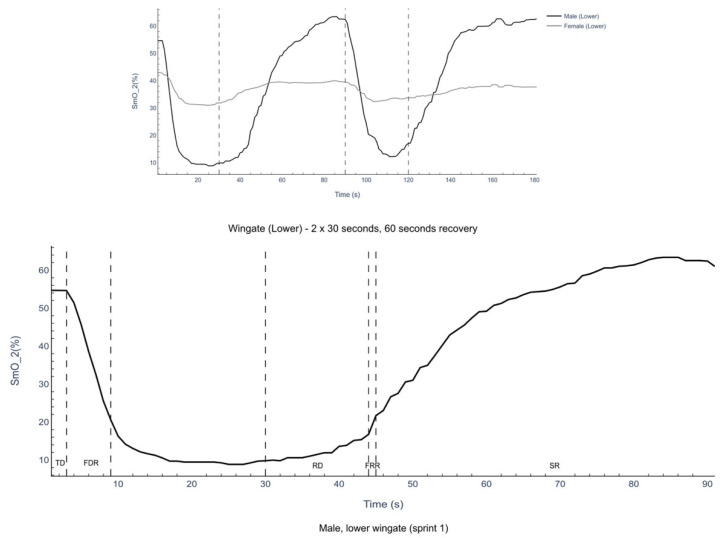
Muscle oxygenation across Wingate-based sprints for males and females. TD = time delay, FRD = fast desaturation rate, SS = steady state RD resaturation delay, FRR = fast resaturation rate, SR = slow recovery. The dotted line reflects the time band for each segment described.

**Figure 2 muscles-04-00018-f002:**
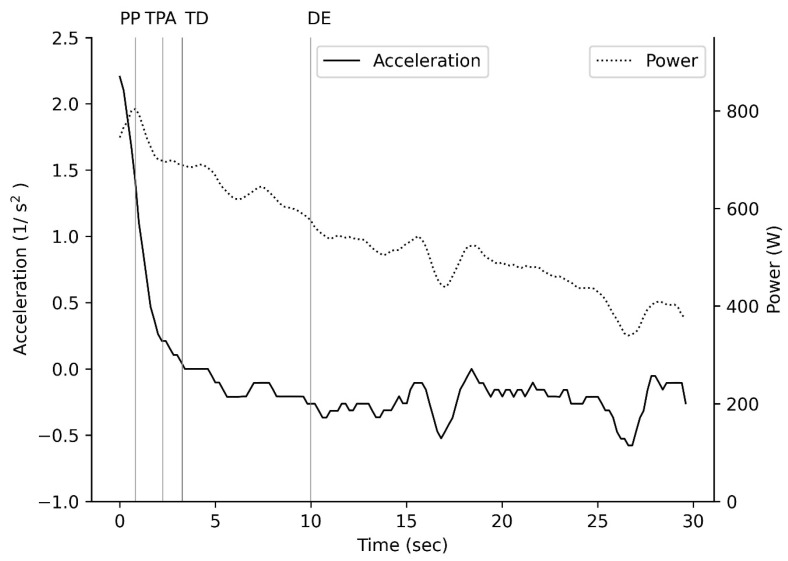
Acceleration and Power across a Wingate-based sprint. PP peak power; TPA time of positive acceleration, TD time delay; DE end fast desaturation.

**Figure 3 muscles-04-00018-f003:**
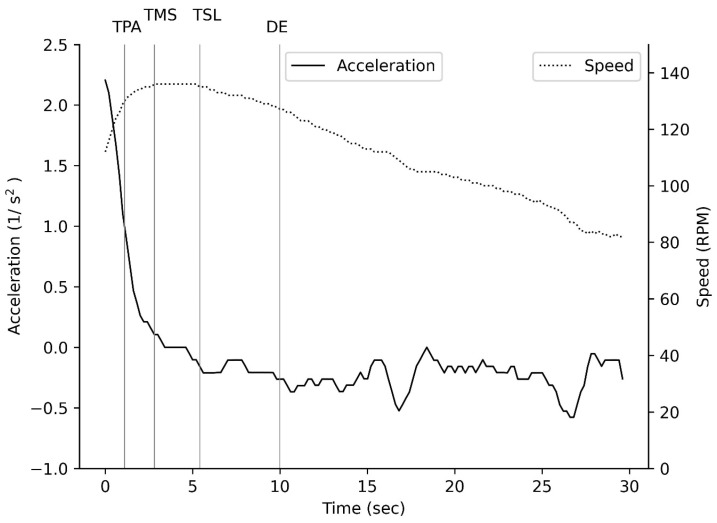
Acceleration and speed across a Wingate-based sprint. TPA time of positive acceleration; TMS time to maximum speed; TSL time to speed loss; DE end fast desaturation.

**Table 1 muscles-04-00018-t001:** Performance during Wingate repeated sprint test. * *p* < 0.05 sprint 1 compared to sprint 2, ⴕ *p* < 0.05 male compared to female for the same sprint. Bold effect size reflects reliable effect based on CI.

					Hedges G Effect Size (95% CIs)			
	Male		Female		Male	Female	Sprint 1	Sprint 2
	Sprint 1	Sprint 2	Sprint 1	Sprint 2	S1 vs. S2	S1 vs. S2	M vs. F	M vs. F
PP (W·kg^−1^)	12.5 ± 1.5 ⴕ	9.6 ± 1.0 ⴕ*	9.7 ± 1.2	7.1 ± 0.6 *	**2.1 (1.0:3.2)**	**2.5 (1.0:4.0)**	**1.9 (0.7:3.1)**	**2.6 (1.3:4.0)**
AP (W·kg^−1^)	8.7 ± 0.6 ⴕ	6.7 ± 0.6 ⴕ*	6.1 ± 1.0	4.8 ± 0.4 *	**3.2 (1.9:4.5)**	**1.9 (0.5:3.2)**	**2.8 (1.4:4.2)**	**3.5 (1.9:5.1)**
MinP (W·kg^−1^)	5.2 ± 1.9	4.2 ± 1.2 *	4.6 ± 0.8	3.4 ± 0.7 *	0.6 (−0.3:1.6)	**1.5 (0.2:2.7)**	0.4 (−0.7:1.4)	0.7 (−0.3:1.8)
PP (W·kg.ffm^−1^)	14.3 ± 2.1	10.9 ± 1.1 ⴕ*	12.6 ± 1.3	9.3 ± 0.9 *	**1.9 (0.9:3.0)**	**2.6 (1.1:4.2)**	0.9 (−0.2:1.9)	**1.4 (0.3:2.6)**
AP (W·kg.ffm^−1^)	9.9 ± 1.1 ⴕ	7.6 ± 0.9 ⴕ*	8.3 ± 1.1	6.3 ± 0.4 *	**2.3 (1.1:3.4)**	**2.2 (0.8:3.6)**	**1.4 (0.3:2.5)**	**1.6 (0.5:2.9)**
MinP (W·kg.ffm^−1^)	6.5 ± 2.0	5.1 ± 1.2 *	6.0 ± 0.9	4.4 ± 0.9 *	0.8 (−0.1:1.7)	**1.6 (0.3:2.8)**	0.3 (−0.7:1.3)	0.5 (−0.5:1.5)
Time to PP (s)	1.3 ± 0.6	3.2 ± 3.7	1.0 ± 0.2	1.7 ± 1.5	−0.7 (−1.6:0.3)	−0.6 (−1.8:0.5)	0.6 (−0.4:1.7)	0.4 (−0.6:1.5)
Time positive acceleration (s)	2.5 ± 1.0	2.6 ± 0.9	2.3 ± 0.4	2.5 ± 1.1	−0.1 (−1.0:0.8)	−0.2 (−1.4:0.9)	0.3 (−0.8:1.26)	0.1 (−0.9:1.1)
Max speed (rpm)	136 ± 9	112 ± 9 *	135 ± 10	123 ± 16	**2.5 (1.3:3.7)**	0.8 (−0.4:2.0)	0.1 (−0.9:1.1)	−0.9 (−1.9:0.2)
Average speed (rpm)	116 ± 7 ⴕ	89 ± 7 ⴕ*	93 ± 14	74 ± 7 *	**3.7 (2.2:5.1)**	**1.6 (0.3:2.9)**	**2.2 (0.9:3.4)**	**2.1 (0.8:3.3)**
Time to max speed (s)	3.6 ± 1.2 ⴕ	3.1 ± 1.6 ⴕ	1.3 ± 0.3	1.1 ± 0.4	0.3 (−0.5:1.2)	0.6 (−0.6:1.7)	**2.3 (1.0:3.6)**	**1.4 (0.3:2.6)**
Time to speed loss (s)	6.0 ± 1.5 ⴕ	4.4 ± 1.9 ⴕ*	2.5 ± 0.3	2.3 ± 0.3	0.9 (−0.1:1.8)	0.6 (−0.6:1.8)	**2.7 (1.3:4.0)**	**1.3 (0.2:2.4)**
Time spent at max speed (s)	2.4 ± 0.9 ⴕ	1.3 ± 0.5 *	1.2 ± 0.2	1.2 ± 0.2	**1.4 (0.4:2.4)**	−0.1 (−1.2:1.0)	**1.5 (0.4:2.6)**	0.1 (−0.9:1.0)

**Table 2 muscles-04-00018-t002:** Muscle oxygen response during repeated Wingate-based sprints. * *p* < 0.05 sprint 1 compared to sprint 2, ⴕ *p* < 0.05 male compared to female for the same sprint.

					Hedges G Effect Size (95% CI’s)			
	Male		Female		Male	Female	Sprint 1	Sprint 2
	Sprint 1	Sprint 2	Sprint 1	Sprint 2	S1 vs. S2	S1 vs. S2	M vs. F	M vs. F
**Left Rec Fem**								
TD (s)	3.2 ± 2.2	3.3 ± 2.8	3.0 ± 2.5	3.0 ± 2.5	−0.0 (−0.9:0.9)	−0.0 (−1.1:1.1)	0.1 (−0.9:1.1)	0.1 (−0.9:1.1)
FDR (%.s^−1^)	−5.6 ± 1.2 ⴕ	−4.2 ± 1.1 ⴕ*	−2.3 ± 1.9	−1.5 ± 0.9	**−1.2 (−2.1: −0.2)**	−0.5 (−1.6:0.7)	**−2.0 (−3.2: −0.8)**	**−2.4 (−3.7: −1.1)**
DD (s)	6.1 ± 1.3 ⴕ	7.3 ± 1.6 ⴕ*	3.7 ± 2.8	4.5 ± 1.0	−0.8 (−1.7:0.1)	−0.3 (−1.5:0.8)	**1.2 (0.1:2.3)**	**1.9 (0.7:3.1)**
Starting SmO_2_ (%)	55.7 ± 10.3	60.0 ± 6.6	50.5 ± 23.3	48.0 ± 18.7	−0.5 (−1.4:0.4)	0.1 (−1.1:1.3)	0.3 (−0.8:1.4)	0.9 (−0.1:2.0)
SS SmO_2_ (%)	17.7 ± 7.5 ⴕ*	25.1 ± 8.0	39.0 ± 20.5	38.4 ± 14.9	−0.9 (−1.9:0.0)	0.0 (−1.2:1.2)	**−1.5 (−2.8:−0.3)**	**−1.2 (−2.2:−0.1)**
** *Recovery* **								
TD (s)	11.0 ± 8.4	8.7 ± 6.0	14.7 ± 12.1	11.7 ± 3.2	0.3 (−0.6:1.2)	0.3 (−0.8:1.5)	−0.4 (−1.4:0.8)	−0.5 (−1.6:0.3)
FRR (%.s^−1^)	2.1 ± 1.0	2.6 ± 1.2 ⴕ	1.4 ± 1.5	0.8 ± 0.6	−0.4 (−1.3:0.5)	0.5 (−0.7:1.6)	0.6 (−0.4:1.6)	**1.6 (0.5:2.8)**
RD (s)	7.7 ± 2.5	6.2 ± 1.9	5.6 ± 2.9	6.6 ± 4.7	0.7 (−0.2:1.6)	−0.2 (−1.4:0.9)	0.8 (−0.3:1.8)	−0.1 (−1.1:0.9)
**Right Rec Fem**								
TD (s)	2.7 ± 1.5	2.7 ± 2.9	3.0 ± 3.4	4.2 ± 2.6	−0.0 (−0.9:0.9)	−0.4 (−1.5:0.8)	−0.1 (−1.1:0.9)	−0.5 (−1.5:0.5)
FDR (%.s^−1^)	−5.5 ± 2.1 ⴕ	−4.5 ± 2.1 ⴕ*	−2.3 ± 1.8	−2.2 ± 2.4	−0.4 (−1.3:0.5)	−0.0 (−1.2:1.2)	**−1.5 (−2.7:−0.3)**	−1.0 (−2.1:0.1)
DD (s)	6.2 ± 0.8 ⴕ	6.6 ± 1.4 ⴕ	3.2 ± 1.8	3.3 ± 2.2	−0.3 (−1.2:0.5)	−0.0 (−1.1:1.1)	**2.2 (1.0:3.5)**	**1.8 (0.6:3.0)**
Starting SmO_2_ (%)	55.3 ± 10.4	54.4 ± 7.4	53.2 ± 14.2	48.8 ± 13.9	0.1 (−0.8:1.0)	0.3 (−0.9:1.4)	0.1 (−0.9:1.1)	0.4 (−0.6:1.5)
SS SmO_2_ (%)	17.6 ± 10.0 ⴕ*	22.8 ± 10.7 ⴕ	38.2 ± 14.7	36.3 ± 10.9	−0.4 (−1.3:0.5)	0.1 (−1.0:1.3)	**−1.7 (−2.9:−0.5)**	**−1.3 (−2.4:−0.2)**
** *Recovery* **								
TD (s)	10.4 ± 7.1	10.7 ± 5.4	12.4 ± 13.4	11.5 ± 6.0	−0.1 (−0.9:0.8)	0.1 (−1.2:1.3)	−0.2 (−1.3:0.9)	−0.1 (−1.2:0.9)
FRR (%.s^−1^)	1.7 ± 0.8	2.0 ± 0.7	0.8 ± 0.7	1.5 ± 0.9	−0.3 (−1.2:0.6)	−0.7 (−2.0:0.6)	1.1 (−0.1:2.2)	0.6 (−0.5:1.7)
RD (s)	9.2 ± 4.1 ⴕ	8.2 ± 5.0	4.8 ± 2.6	4.7 ± 3.0	0.2 (−0.7:1.1)	0.0 (−1.2:1.3)	1.1 (−0.0:2.3)	0.7 (−0.4:1.8)

## Data Availability

Data is available upon reasonable request, by contacting the corresponding author.

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
