# Peer review of "The Importance of Skeletal Muscle Oxygenation Kinetics for Repeated Wingate-Based Sprint Performance"

_muscles, 2025, doi:10.3390/muscles4020018_

Round 1
Reviewer 1 Report
Comments and Suggestions for Authors
Dear authors: Congratulations on your paper, which appears to be a continuation of the work published in 2023 (the same 10 male subjects).
The work is well developed, the objective is clear, and the methodology and analysis system are appropriate for this type of research. The materials used are current and accessible to other researchers. The results are well presented, and the graphs are very illustrative. The discussion provides supplementary information, and the conclusions are in line with the results. I agree with you regarding the study's limitations (control of training load; the use of the Lode ergometer versus the Monark) and also the lack of a control group. However, I believe that, in the future, your data may be useful to improve training for this type of athlete.
I wish you the best of luck.
Author Response
Thank you for taking the time to review our manuscript and your comments.
Reviewer 2 Report
Comments and Suggestions for Authors
Abstract
Poorly written or ambiguous sentences:
- “Mean power (-0.615) and average velocity (-0.564) were correlated…”: Unclear. It is not evident whether this refers to correlation coefficients or to another metric.
- The abstract lacks key methodological information (e.g., number of participants, NIRS technology, structure of the protocol).
- It is not clear what this study contributes that is genuinely novel compared to existing literature.
Introduction
- Absence of an explicit hypothesis: The authors do not clearly state a hypothesis such as: "We expect that male boxers will present faster oxygenation kinetics than females," for example.
Contradictory citations without clarification:
- Two studies are cited (Ghosh et al., 1995 and Davis et al., 2014) with opposing estimates of boxing’s energy system profile (80% anaerobic vs. 80% aerobic), but no explanation is provided to reconcile this discrepancy.
- Unjustified speculation:
- The statement: "power may not be a good metric of muscular demand during a Wingate test due to the inefficiency of energy transfer..." (lines 64–65), is made without discussing alternatives or providing direct evidence that this is the case in their sample.
- Redundancy: The notion that the Wingate test reflects the physiological demands of boxing is repeated several times without offering new justification.
Methods
- Very small sample size (n = 16; 10 males and 6 females). No power analysis is reported to justify the sample size.
- No control for training cycle phase or menstrual cycle in women, which may affect muscle oxygenation—particularly relevant if sex-based comparisons are to be made.
- Selection bias: Participants were included based on availability rather than homogeneous criteria (line 366), which could introduce uncontrolled variability.
- Lack of clarity in NIRS data processing: It is not explained whether motion artifacts were corrected or how steady state was defined.
- The variable “fold activity” is newly created by the authors without prior validation. Its limitations or potential sensitivity to noise are not discussed.
Results
- Inconsistencies in table formatting: Table 1 is excessively large and difficult to read. Many important comparisons lack 95% confidence intervals.
- Confusing interpretation at times: For instance, differences between men and women in certain variables are mentioned without clarifying whether they are statistically significant (e.g., "peak velocity", lines 194–197).
- Lack of multivariate analysis or consistent adjustment for fat-free mass across all comparisons.
- Some correlations reported are weak (e.g., -0.54), yet they are discussed as if they were strong.
Discussion
- Speculative statements without direct experimental support:
- “This may be due to the impact of oral contraceptives on estrogen and mitochondrial function…” (line 321): this was not evaluated in the study, nor was contraceptive use among participants recorded. Therefore, this is a speculative hypothesis with no empirical support in this paper.
- “Women should train at higher intensities than men” (line 392): a risky and overly generalized claim that overlooks the principle of training individualization.
- The impact of small sample size on external validity and the risk of Type II errors is not adequately discussed.
- Repeated Wingate is presented as a novel recovery assessment tool, yet prior studies in sports like judo, cycling, and football have used similar protocols (some cited in the introduction).
Conclusions
- Conclusions are overly general and assertive for a study with so many limitations.
- Example: “The Wingate test is not anaerobic” (line 388). This statement disregards decades of validation of the Wingate test as a predominantly anaerobic measure. A more accurate statement would be: “an appreciable aerobic component emerges within the first few seconds,” but the test’s anaerobic foundation should not be denied.

Author Response
Thank you very much for taking the time to review this manuscript. Please find the detailed responses below and all amendments have been marked in red in the modified manuscript.
Review: The importance of skeletal muscle oxygenation kinetics for repeated Wingate performance
Abstract Poorly written or ambiguous sentences:
A background sentence has been added to improve clarity of abstract
- “Mean power (-0.615) and average velocity (-0.564) were correlated…”: Unclear. It is not evident whether this refers to correlation coefficients or to another metric.
- The abstract lacks key methodological information (e.g., number of participants, NIRS technology, structure of the protocol).
The number of male and female participants has been stated in the abstract, it is unusual to name the technology used in the abstract but this has been added, the full structure has been presented but this has been rewritten for clarity.
- It is not clear what this study contributes that is genuinely novel compared to existing literature.
The abstract and highlights have been amended to highlight the novelty of the findings more.
Introduction
- Absence of an explicit hypothesis: The authors do not clearly state a hypothesis such as: "We expect that male boxers will present faster oxygenation kinetics than females," for example.
Hypothesis has been added - It was hypothesised that there would be a decrease in power across the 2 sprint efforts and this would be related to the ability of the rectus femoris muscle to recover from the initial sprint effort. Further it was hypothesised that the response to repeated Wingate based sprint efforts would be different in male and female boxers.
Contradictory citations without clarification:
- Two studies are cited (Ghosh et al., 1995 and Davis et al., 2014) with opposing estimates of boxing’s energy system profile (80% anaerobic vs. 80% aerobic), but no explanation is provided to reconcile this discrepancy.
Explanation added - This may reflect differences in the workrate of the bouts, with Ghosh having an activity frequency of 1.7-2.0/s and Davies having an activity frequency of 1.2/s.
- Unjustified speculation:
- The statement: "power may not be a good metric of muscular demand during a Wingate test due to the inefficiency of energy transfer..." (lines 64–65), is made without discussing alternatives or providing direct evidence that this is the case in their sample.
This has been updated - However, power may not be a good metric of muscular demand during a Wingate test due to the inefficiency of transfer of energy from the pedal to the flywheel (Robergs et al., 2015). The kinetic energy produced in the muscle during a cycle stroke is directly related to the pedal rate (Driss and Vandewalle., 2013.). This may mean comparison across studies is incorrect due to differences in ergometer mechanics and cadence metrics should also be reported to allow for better comparison.
- Redundancy: The notion that the Wingate test reflects the physiological demands of boxing is repeated several times without offering new justification.
This section has been deleted - Within the structure of a boxing match there is a need to understand the physiological response of the athlete to repeated high intensity bursts with 1 minute of recovery. Likewise, there is a need to examine the ability of the rectus femoris muscle to perform repeatedly under high physiological load. The use of repeated Wingate based sprints in boxing would allow for a controlled evaluation of the athlete under high load conditions.
Methods
- Very small sample size (n = 16; 10 males and 6 females). No power analysis is reported to justify the sample size.
No power test was carried out prior to the study as access to participants was by convienence. It is inappropriate to carry out a post hoc power test. However to mitigate this we have provided hedges g effect size with 95% confidence intervals in table 1 and 2 and have provided the 95% CI for correlations. When the confidence interval does not pass through zero then the effect or relationship can be assumed to be robust (see for fuller discussion: Dziak JJ, Dierker LC, Abar B. The Interpretation of Statistical Power after the Data have been Gathered. Curr Psychol. 2020 Jun;39(3):870-877. doi: 10.1007/s12144-018-0018-1.) Whilst 6 females is small it represents over 10% of the female professional boxers in the UK. Within the UK there are currently 55 active female professional boxers. For males there are 1045 active professional boxers in the UK, so 10 represents 1% of the overall population. However in Scotland there are only 109 male professional boxers so 10 represents 9% of the available population.
- No control for training cycle phase or menstrual cycle in women, which may affect muscle oxygenation—particularly relevant if sex-based comparisons are to be made.
A sentence has been added to give the training status. Evidence shows that oral contraceptive and menstrual phase has no impact on NIRS derived HHB kinetics during exercise (see Mattu, A.T., Iannetta, D., MacInnis, M.J., Doyle‐Baker, P.K. and Murias, J.M., 2020. Menstrual and oral contraceptive cycle phases do not affect submaximal and maximal exercise responses. Scandinavian journal of medicine & science in sports, 30(3), pp.472-484.). A sentence has been added to the methods.
- Selection bias: Participants were included based on availability rather than homogeneous criteria (line 366), which could introduce uncontrolled variability.
This has been discussed as a potential limitation to the study. I would argue all studies are carried out based on participant availability. If potential participants are not available when testing is being conducted, then they would be excluded. The criterion for participation is clearly stated, the participants had to hold an active BBBC license. This criterion already limited participants eg no amateur boxers could take part, but it does ensure that we are recruiting from a very particular population that is reflective of a professional boxing cohort.
- Lack of clarity in NIRS data processing: It is not explained whether motion artifacts were corrected or how steady state was defined.
This has been amended but details of the method are provided in the referenced article: Briefly, a 5 second median filter was applied to the data to remove movement artefacts prior to linear curve fitting being applied to the data to look at the relationship between SmO2 and time and SmO2 and heart rate. Three clear components in the NIRS signal are present in each sprint: initial time delay where oxygenation is stable, a fast desaturation where oxygenation levels are falling and a new equilibrium where oxygenation is lower but stable (Figure 1). The new equilibrium was defined as the end of the fast linear desaturation until the end of the 30 second period. Two clear linear components are present during recovery: initial time delay where oxygenation is stable, a fast resaturation where oxygenation levels are rising rapidly before going into a non-linear component (figure 1).
- The variable “fold activity” is newly created by the authors without prior validation. Its limitations or potential sensitivity to noise are not discussed.
This has been removed, to allow validation of the concept
Results
- Inconsistencies in table formatting: Table 1 is excessively large and difficult to read. Many important comparisons lack 95% confidence intervals.
Font has been reduced to make all data fit in a single line. 95% CIs are given only for the effect size to show reliability of the finding. Standard deviation is given with the mean values which is the normal way this type of data is presented.
- Confusing interpretation at times: For instance, differences between men and women in certain variables are mentioned without clarifying whether they are statistically significant (e.g., "peak velocity", lines 194–197).
It is unclear where this is as peak velocity is not a term used in the manuscript. However expression of statistics has been updated to provide main effects and then post hoc analysis. Hedges g has also been calculated to allow effect to be seen along with 95% CI to show reliability of effect.
- Lack of multivariate analysis or consistent adjustment for fat-free mass across all comparisons.
Multivariate regression has been carried out for correlative analysis.
- Some correlations reported are weak (e.g., -0.54), yet they are discussed as if they were strong.
I would disagree, if you look at scaling for Pearsons then 0.5 and above are viewed as large associations (see Yagin 2024 https://e-jespar.com/index.php/jespar/article/view/27/31). However the correlative analysis has been completely redone as a multivariate regression.
Discussion
- Speculative statements without direct experimental support:
- “This may be due to the impact of oral contraceptives on estrogen and mitochondrial function…” (line 321): this was not evaluated in the study, nor was contraceptive use among participants recorded. Therefore, this is a speculative hypothesis with no empirical support in this paper.
This has been removed. It was added to offer the reader potential reasons why we see a difference between males and females but accept that it is speculative
- “Women should train at higher intensities than men” (line 392): a risky and overly generalized claim that overlooks the principle of training individualization.
This has been changed to females may need to train differently to males
- The impact of small sample size on external validity and the risk of Type II errors is not adequately discussed.
This has been added
- Repeated Wingate is presented as a novel recovery assessment tool, yet prior studies in sports like judo, cycling, and football have used similar protocols (some cited in the introduction).
However none of these studies have looked at muscle oxygenation recovery and subsequent impact on high intensity performance.

Reviewer 3 Report
Comments and Suggestions for Authors
Dear corresponding author, thank you for submitting your work to the journal Muscles.
Brief Summary
This study explores the responses of muscular oxygenation and anaerobic performance during repeated sprints on a cycle ergometer (Wingate test) in male and female professional boxers. The work analyses sex differences and proposes correlations between oxygen kinetics (NIRS) and performance parameters (power, speed), suggesting practical implications for boxing training.
General Comments
The manuscript addresses a relevant topic for physiology applied to combat sports. The use of NIRS spectroscopy in this context appears interesting and allows to deepen the understanding of muscular oxidative dynamics during high-intensity efforts. However, some methodological aspects should be clarified: the description of the mono-exponential fitting and the concept of "fold activity" should be detailed more. Moreover, the section concerning the author contribution is missing, which is necessary to comply with MDPI guidelines.
Specific Comments
- Line 113–115: clarify whether SmO2 measurement was done bilaterally but analysed as average or separately (even though later it is said that the average value was taken, it is not so clear...).
- Line 122: you used the term Wingate improperly. What reference did you use to justify a Wingate with active recovery when it doesn't exist? This is an important issue and I would like this to be clarified.
- Line 137–138: the description of the linear fitting needs more details; it is not clear if the analysis was based on visually defined segments or objective criteria.
- Missing: the “Author Contributions” section, which is essential for transparency of responsibilities.
- Missing: “Ethics Statement” and “Informed Consent Statement” should be completed, even if informed consent is vaguely mentioned.
I believe the topic is current, but not the approach to the problem. On my opinion, you simplified and generalised using the tests in an improper way. The work in this form is not publishable, but if the authors can provide more correct information I will read an improved version.
Author Response
Thank you very much for taking the time to review this manuscript. Please find the detailed responses below and all amendments have been marked in red in the modified manuscript.
Brief Summary
This study explores the responses of muscular oxygenation and anaerobic performance during repeated sprints on a cycle ergometer (Wingate test) in male and female professional boxers. The work analyses sex differences and proposes correlations between oxygen kinetics (NIRS) and performance parameters (power, speed), suggesting practical implications for boxing training.
General Comments
The manuscript addresses a relevant topic for physiology applied to combat sports. The use of NIRS spectroscopy in this context appears interesting and allows to deepen the understanding of muscular oxidative dynamics during high-intensity efforts. However, some methodological aspects should be clarified: the description of the mono-exponential fitting and the concept of "fold activity" should be detailed more. Moreover, the section concerning the author contribution is missing, which is necessary to comply with MDPI guidelines.
Fold activity has been removed to allow this concept to be validated fully prior to using. Author contributions are provided at the end of the manuscript. Author Contributions: Investigation and data collection, A.U. and J.B.; Writing – original draft, A.U. and J.B.; Writing – review & editing, A.U. and J.B.
Specific Comments
- Line 113–115: clarify whether SmO2 measurement was done bilaterally but analysed as average or separately (even though later it is said that the average value was taken, it is not so clear...).
- This has been clarified and the correlation analysis been redone to look at components with regards each leg
- Line 122: you used the term Wingate improperly. What reference did you use to justify a Wingate with active recovery when it doesn't exist? This is an important issue and I would like this to be clarified.
- The terminology has been corrected to Wingate based sprints throughout the manuscript. This is to help define the resistance of the 2 x 30s sprints that have been performed.
- Line 137–138: the description of the linear fitting needs more details; it is not clear if the analysis was based on visually defined segments or objective criteria.
Linear segments were identified based on the point of rapid SmO₂ desaturation, typically occurring near peak power output. Segments were selected using both visual inspection and confirmation via high R² values (>0.90) from the linear regression fit,I have attached an example graph.
- Missing: the “Author Contributions” section, which is essential for transparency of responsibilities.
- This is presented at the end of the manuscript Author Contributions: Investigation and data collection, A.U. and J.B.; Writing – original draft, A.U. and J.B.; Writing – review & editing, A.U. and J.B.
- Missing: “Ethics Statement” and “Informed Consent Statement” should be completed, even if informed consent is vaguely mentioned.
- This has been provided at the end of the manuscript –
Institutional Review Board Statement: The study was conducted in accordance with the Declaration of Helsinki, and approved by the Ethics Committee of Abertay University (protocol code EMS4768 and EMS7056, approved on 10 November 2021 and 3 Feb-ruary 2023).
Informed Consent Statement: Informed consent was obtained from all subjects involved in the study.

Round 2
Reviewer 2 Report
Comments and Suggestions for Authors
The manuscript presents rigorous and novel research on the kinetics of skeletal muscle oxygenation in professional boxers, with an appropriate experimental design and sound statistical treatment. Significant improvements have been made compared to previous versions, especially in terms of methodological clarity, multivariate analysis, and scientific discussion.
However, I recommend the following minor revisions, which aim to improve the clarity and practical applicability of the work:
Terminological consistency:
The manuscript uses several terms to refer to muscle oxygenation metrics, such as “steady-state SmO₂,” “equilibrium SmO₂,” and “final SmO₂,” without clearly indicating whether they are equivalent. Please standardize the terminology or briefly clarify their differences in the methods section.
Asymmetry in the response of the legs:
Given that more significant associations are observed in the left leg, it would be advisable to include a brief comment in the discussion proposing a hypothesis or justification for this asymmetry, or to acknowledge it as a limitation.
Wording in the methods section:
Some expressions show inconsistencies in tense or use an informal tone (e.g., lines 135-137). Please unify the writing style to maintain a formal scientific tone throughout the methods section.
Practical applicability:
In the “Perspective” section, it would be useful to include a brief statement on how coaches or professionals could apply the results (e.g., using desaturation/resaturation patterns as indicators of athletes' readiness or recovery capacity).
Minor punctuation and repetition issues:
In the results section, the phrase “no significant relationships were observed with individual components...” is repeated verbatim several times. Consider summarizing these results to improve narrative flow.
Once these minor revisions are implemented, I believe the manuscript will be ready for publication. I commend the authors for the innovative approach of the study and the substantial improvements in this version.
Author Response
Thank you once again for your comments, the amendments have been highlighted in yellow.

Reviewer 3 Report
Comments and Suggestions for Authors
I have carefully read the modifications made by the authors. I believe the work still has some criticalities but they are not resolvable in this study. Balancing the pros and cons, I think it can be accepted in this form.
Author Response
We would like to thank for your comments.